# REN: Fast and Efficient Region Encodings from Patch-Based Image Encoders

**Savya Khosla**    **Sethuraman TV**    **Barnett Lee**    **Alexander Schwing**    **Derek Hoiem**

University of Illinois Urbana-Champaign

{savyak2,st34,bl29,aschwing,dhoiem}@illinois.edu

## Abstract

We introduce the *Region Encoder Network (REN)*, a fast and effective model for generating region-based image representations using point prompts. Recent methods combine class-agnostic segmenters (e.g., SAM) with patch-based image encoders (e.g., DINO) to produce compact and effective region representations, but they suffer from high computational cost due to the segmentation step. REN bypasses this bottleneck using a lightweight module that directly generates region tokens, enabling *60× faster token generation* with *35× less memory*, while also *improving token quality*. It uses a few cross-attention blocks that take point prompts as queries and features from a patch-based image encoder as keys and values to produce region tokens that correspond to the prompted objects. We train REN with three popular encoders—DINO, DINOv2, and OpenCLIP—and show that it can be extended to other encoders without dedicated training. We evaluate REN on semantic segmentation and retrieval tasks, where it consistently outperforms the original encoders in both performance and compactness, and matches or exceeds SAM-based region methods while being significantly faster. Notably, REN achieves state-of-the-art results on the challenging Ego4D VQ2D benchmark and outperforms proprietary LMMs on Visual Haystacks' single-needle challenge. The code and pretrained models are available at `https://github.com/savya08/ren`.

## 1 Introduction

Do we really need hundreds of patch-based tokens to understand every image? Existing ViT encoders divide images into a fixed grid of patch tokens. This approach is inefficient for two main reasons: (1) images with few objects are represented using hundreds of tokens, leading to unnecessarily high memory and compute costs for applications like video understanding; and (2) patch tokens do not align with object boundaries, making tasks like retrieval and segmentation more challenging. A more scalable and effective alternative is a region-based representation, which divides images into object-level tokens and represents simpler images using far fewer tokens.

Recent region-based representations average-pool patch features over regions generated by class-agnostic segmentation methods, such as SAM. These approaches improve semantic segmentation and retrieval accuracy, but generating accurate segmentation masks for all objects in an image remains computationally expensive, with high-quality segmentation models requiring up to 30× more time and memory than patch-based image encoders. This significantly limits the practical usability of region-based representations. Furthermore, existing SAM-based methods for region token generation rely on simple linear aggregation of patch features within object masks, which can remove fine-grained details and discard valuable context surrounding objects. Current segmentation models also tend to produce overlapping masks or miss parts of an image, resulting in over-representation of some regions and no representation for others.

39th Conference on Neural Information Processing Systems (NeurIPS 2025).

To address these limitations, we propose the Region Encoder Network (REN), a point promptable model that **transforms patch-based features into region-based representations without requiring explicit segmentation**. The key idea is that image features already contain sufficient information to segment objects. So instead of relying on external segmentation masks, we train REN to implicitly segment and pool information through attention. Specifically, as shown in Figure 1, REN uses a lightweight cross-attention module trained with two learning objectives—contrastive learning and feature similarity—to extract high-quality region tokens directly from patch features. Given a point prompt (query), REN applies cross-attention over patch features (keys/values) to produce a token representing the object corresponding to the queried point. To generate region tokens for all objects, REN can be prompted with a grid of points followed by a simple token aggregation step to merge tokens that correspond to the same object.

We explore the design choices that make REN effective and apply it to multiple patch-based image encoders—DINO [5], DINOv2 [32], and OpenCLIP [17]. Our experiments show that REN consistently outperforms the patch-based feature backbone on semantic segmentation and object retrieval. Further, it performs comparably to SAM-based approaches, while being more than **60× faster** and while using **35× less memory**. Our main contributions can be summarized as follows:

- We introduce REN, a point promptable model that efficiently transforms patch-based feature maps into semantically meaningful region tokens using a lightweight cross-attention module, eliminating the need for costly segmentation models. We show that while REN can be trained for any patch-based encoder, we can also extend it to other encoders without dedicated training.

- We study the design choices that affect the effectiveness and efficiency of transforming patch-based features into region tokens. For instance, we investigate the impact of prompting strategies, how to guide attention mechanisms effectively, how to preserve the information encoded in backbone features, and how to move beyond simple pooling operations for feature aggregation—such as average or max pooling—used in prior works.

- We train REN with multiple image encoders and will release the code and trained models to facilitate future research and applications.

## 2   Related Works

**Patch-Based Image Encoders.** The Vision Transformer [9] introduced the paradigm of representing images as sequences of fixed-size patch tokens processed by Transformers [43]. Building on this, numerous patch-based image encoders have been developed to learn rich visual semantics [5, 32, 14, 54, 2] or align vision with other modalities like text [33, 17, 50, 41, 53]. These backbones power a wide range of applications—for example, SAM [22] uses MAE [14], and LLaVA [25] builds on CLIP [33]. However, they represent each image with hundreds of tokens, regardless of content, leading to high computational overhead. Techniques like token pruning [34, 26, 49, 31] and merging [28, 23, 47, 37, 30, 27] aim to reduce token count, but they often trade off performance and still rely on grid-based, semantically weak tokens. This motivates exploring beyond patch tokens.

**Learning Objectness.** The notion of objectness—that certain regions in an image correspond to coherent, meaningful entities—has inspired many works. Self-supervised methods like LOST [40] and TokenCut [45] aim to segment objects by clustering image patches based on feature affinity. In parallel, supervised models like DETR [4] and MaskFormer [8, 7] learn to recognize objects using cross-attention between learnable queries and patch features under dense supervision. More recently, SAM and its variants [22, 35, 55] have demonstrated that sparse prompts (e.g., points) can effectively guide segmentation, showing that rich object-level information can emerge from simple supervision signals. These works demonstrate that patch-level features can support object-level understanding. Building on this, we propose a lightweight module that extracts generic object representations from frozen patch features—without complex objectives or extensive supervision.

**Region-Based Representations.** Region-based representations have a long history in computer vision, where regions were traditionally defined using low-level cues such as color, texture, and intensity [12, 15, 16, 29, 36]. However, these early segmentation methods lacked precision. The development of powerful class-agnostic segmenters like SAM [22] has reignited interest in region-based representations [38]. These methods use region masks from SAM to average-pool features from patch-based encoders, producing region tokens that outperform patch tokens on tasks like segmentation and retrieval, while significantly reducing token count per image. Follow-up works further demonstrate their effectiveness in instance localization [20], embodied navigation [11], long-

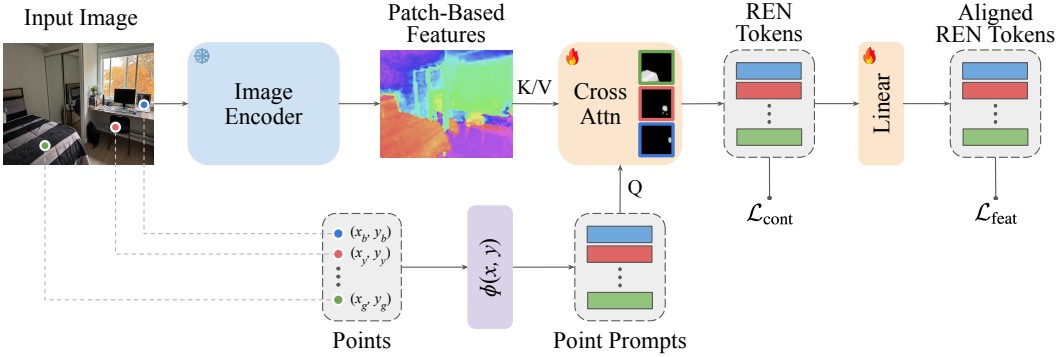

(a) REN Architecture

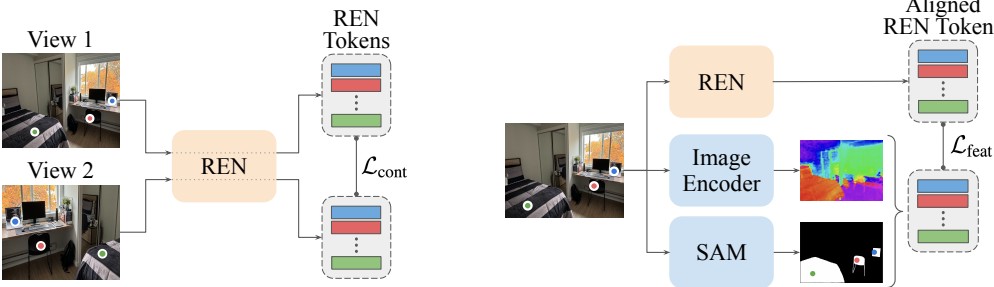

(b) Training Objectives: Contrastive Token Learning and Feature Similarity

Figure 1: **Overview of REN.** Point prompts interact with patch-based features through cross-attention blocks to produce region tokens. The training objective combines two components: (1) a contrastive loss that aligns region tokens with those generated from an augmented view of the same image, and (2) a feature similarity loss that aligns a linear projection of these tokens with average-pooled patch features obtained using SAM masks. REN eliminates the need for explicit segmentation at inference time while producing efficient and semantically rich region representations. We also show thresholded attention maps for three query points inside the cross-attention block, which show that the model learns to aggregate features primarily from the regions marked by the corresponding point prompts.

tail object search [39], and multimodal concept learning [3]. While these SAM-based methods have revived region-based representations, use remains limited due to high computational cost, coarse feature aggregation, and incomplete image coverage. In contrast, REN uses lightweight cross-attention from point prompts to patch features, enabling fast and memory-efficient region token generation. This learned aggregation captures fine-grained, context-aware information and ensures full image coverage with a compact set of region tokens. As a result, REN improves both efficiency and performance across tasks while adapting the token count to image complexity.

## 3 REN

Figure 1 provides an overview of the proposed method. REN generates region tokens by cross-attending features from a frozen image encoder with point prompts, where each point prompt is represented using a 2D sinusoidal position embedding. We use a series of four cross-attention blocks. The keys and values for each block are a linear projection of the image encoder features. The queries for the first block are generated using a linear projection of the point prompts. For the subsequent blocks, the queries are generated using a sum of point prompts and the previous block's output. The final block outputs region tokens for the queried points.

### 3.1 Training via Self-Supervised Learning and Knowledge Retention

We train REN using images from the Ego4D dataset [13], which offers diverse scenes with objects at varying scales and clutter. For each image encoder, a single REN model is trained and used across

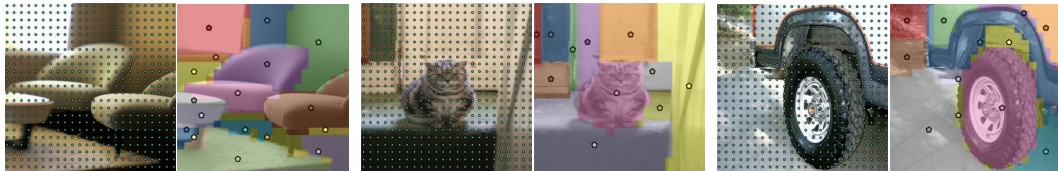

(a) Image pairs showing grid-based point prompts (left) and token aggregated points (right).

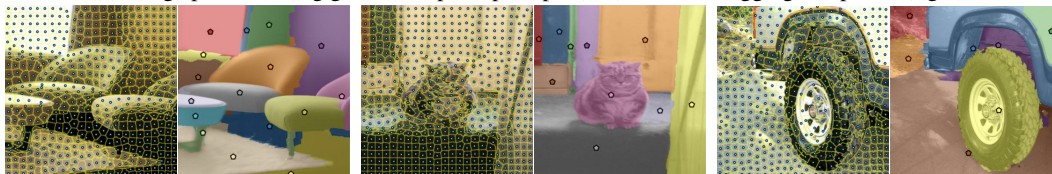

(b) Image pairs showing SLIC-based point prompts (left) and token aggregated points (right).

Figure 2: **Point prompting strategies and token aggregation results.** Region tokens corresponding to point prompts within the same-colored area are aggregated, and we show a representative point prompt for each region. Thus, each image can be represented with a few dozen tokens instead of the hundreds required by patch-based methods. (Best viewed in color)

all tasks. We preprocess the training dataset to extract segmentation masks using SAM [22]. These masks are used to guide the contrastive and feature similarity losses, as described below.

**Contrastive Token Learning.** We use the InfoNCE loss [42] to ensure region tokens are consistent across different views of the same object. Specifically, we identify regions in the training images using SAM [22] and assign an ID to each region. Then, we generate two augmented views of each image using random horizontal flipping, rotation, cropping, color jitter, and sharpness adjustments. REN is used to extract region tokens from both views. These "REN tokens" corresponding to prompt points that fall inside the same region (i.e., are fully contained within the same SAM-generated mask) are treated as positives, while all others are considered negatives. Specifically, we compute

$$\mathcal{L}_{\text{cont}} = -\frac{1}{N} \sum_{i=1}^{N} \log \frac{\sum_{j=1}^{N} \mathbb{1}_{[j \neq i, \ \text{id}_j = \text{id}_i]} \exp(\text{sim}(r_i, r_j)/\tau)}{\sum_{k=1}^{N} \mathbb{1}_{[k \neq i]} \exp(\text{sim}(r_i, r_k)/\tau)}, \tag{1}$$

where $r_i$ denotes the $i^{\text{th}}$ REN token, $\text{id}_i$ denotes the region ID of the $i^{\text{th}}$ token, $N$ is the number of region tokens in the image pair, $\tau$ is a temperature hyperparameter set to 0.1, $\text{sim}(\cdot)$ represents the cosine similarity function, and $\mathbb{1}_{[\cdot]}$ is an indicator function.

**Feature Similarity.** Training only on the contrastive loss could cause the model to drift away from the original encoder's feature space. This is undesirable, as the original models are well-trained using large and carefully curated datasets, and we would like to retain their generalization capability. To this end, we introduce a feature similarity objective that encourages alignment between region tokens and the original encoder features. Specifically, for each region, we generate a target token $t_i$ by average pooling encoder features within the SAM mask [38], and apply cosine embedding loss against the "aligned REN token" $\tilde{r}_i$, obtained via a linear projection of the REN token $r_i$:

$$\mathcal{L}_{\text{feat}} = \frac{1}{N} \sum_{i=1}^{N} \left( 1 - \frac{t_i \cdot \tilde{r}_i}{\|t_i\| \|\tilde{r}_i\|} \right). \tag{2}$$

The linear projection is crucial because it allows the contrastive objective to flexibly learn more discriminative region tokens, without being overly constrained by the structure of the original feature space. At the same time, this objective ensures that the generated region representations retain information to approximate the average backbone features via a simple linear projection.

**Overall Loss.** REN is trained using the following overall loss: $\mathcal{L} = \lambda_{\text{cont}} \mathcal{L}_{\text{cont}} + \lambda_{\text{feat}} \mathcal{L}_{\text{feat}}$. We set $\lambda_{\text{cont}} = \lambda_{\text{feat}} = 1.0$.

## 3.2 Inference via Prompting and Token Aggregation

Inference is straightforward when the target object is specified by the user or the task annotation. E.g., in object retrieval, a mask is provided for the query object, and REN can be prompted with points on the annotated object mask to get region tokens for the query. However, for tasks requiring region tokens for all objects, parts, and background, REN must be prompted with multiple point prompts across the image, followed by token aggregation to merge tokens that correspond to the same object.

**Point Prompting.** We present two ways of prompting REN, as illustrated in Figure 2: (1) Grid-based prompting, which uses a $G \times G$ grid of points as prompts; and (2) SLIC-based prompting, where we segment the image into $S = G^2$ superpixels using SLIC [1] and use the center of each superpixel as a point prompt. Grid-based prompting incurs no computational cost but may produce a few noisy tokens from prompts at object boundaries. SLIC-based prompting eliminates ambiguous prompts at the cost of an inexpensive grouping step. We use Fast-SLIC [21] to get SLIC-based prompts.

**Token Aggregation.** We aggregate tokens by first computing the pairwise cosine similarity between all REN tokens. Based on this similarity, we construct an adjacency matrix where edges are formed for pairs with a similarity greater than $\mu = 0.975$. Connected components in this graph are identified, and tokens within each component are average pooled to produce a single representative token. When prompts are densely sampled (i.e., with high $G \times G$ or $S$), some points fall on object boundaries, producing noisy region tokens that fail to group with others. These outlier tokens can be safely discarded without impacting downstream performance. In particular, when using dense point prompts (e.g., over 1000 points), we discard all groups that contain fewer than three tokens. Figure 2 shows the output after token aggregation, demonstrating that this step significantly reduces the number of tokens required to represent an image.

**Variants of Region Representations.** As mentioned in Section 3.1, REN produces two sets of token-level representations: (1) *REN tokens*, which are directly optimized using the contrastive loss (Equation (1)), and (2) *Aligned REN tokens*, which are trained to align with the original image features via the feature similarity loss (Equation (2)). Since aligned REN tokens can be derived via a linear projection of the REN tokens, the latter contains all the information needed to obtain the former. As a result, we use REN tokens for downstream tasks that involve further learning (e.g., semantic segmentation). For learning-free setups, the choice depends on the task: REN tokens are better for tasks requiring discriminative representations of the same object (e.g., visual query localization), while aligned REN tokens are preferable for tasks that leverage the properties of the underlying image encoders—for example, CLIP's language alignment (e.g., visual haystacks) or DINOv2's category-level semantic understanding (e.g., image retrieval).

## 3.3 Extending Pretrained REN to Other Encoders

We can use a pretrained REN—for example, REN trained for DINO ($REN_{DINO}$)—to generate region-based representations from any target image encoder without dedicated training. Specifically, given an input image, we use SLIC-based point prompts to generate region tokens from $REN_{DINO}$, and track which point prompts are grouped together during token aggregation. For each group of points, we get a region mask by taking a union of the corresponding superpixels. Figure 2b illustrates examples of such region masks. To extract a feature representation for each region, we use the region masks to constrain the global attention in the final layer of the target image encoder. Specifically, for each region mask, we identify the set of patch tokens that overlap with the region. Then, we restrict the attention of the CLS or query token (used for global aggregation) only to the patch tokens that fall within the region. This can be efficiently implemented by duplicating the CLS/query token and applying attention masking using the region masks. By doing this for each region, we obtain region tokens that align with the feature space of the target encoder.

The REN-based approach offers two key advantages over the SAM-based method proposed by Shlapentokh-Rothman et al. [38]: (1) substantially lower computational cost in both time and memory, and (2) complete coverage of the image, as unlike SAM, the masks produced by REN do not overlap and leave no area unsegmented.

Table 1: **Runtime comparison.** With $10\times$ fewer parameters, REN achieves over $60\times$ speedup compared to the fastest SAM-based approach, as measured on a single NVIDIA A40 GPU. Evaluations use either a $32\times32$ grid prompts or 1024 SLIC-based prompts. Reported metrics exclude the patch-based image encoder (DINO ViT-B/8: 85.8M parameters, 0.011 s/img).

| Method | Params (M) | Time (s/img) | |
|---|---|---|---|
| | | SLIC | Grid |
| SAM [22] | 641.1 | | 2.310 |
| Batched-SAM [24] | 641.1 | | 2.048 |
| EfficientViT-SAM [51] | 203.3 | | 1.790 |
| SAM 2 [35] | 224.4 | | 1.898 |
| REN (w/o TokAgg) | 20.1 | 0.020 | 0.017 |
| REN (w/ TokAgg) | 20.1 | 0.033 | 0.029 |

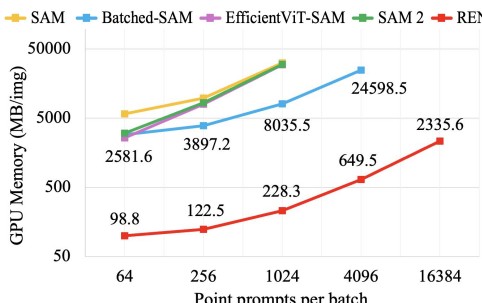

Figure 3: **Memory comparison.** We measure peak GPU memory used for processing a single image across prompt batch sizes. REN can handle larger batches and it uses substantially less memory—for example, $35\times$ less memory than the most efficient SAM-based approach for a batch of 1024 prompts ($32\times32$ grid).

## 4 Experiments

We compare the computational cost of REN and SAM-based methods in Section 4.1, evaluate REN on downstream tasks in Section 4.2, and present ablation studies in Section 4.3. We additionally discuss alternative approaches explored in Section B of the supplementary.

### 4.1 Compute Requirements

We compare the computational requirements of REN with other SAM-based methods for region token generation. This analysis is conducted on a single NVIDIA A40 using images from the ADE20K dataset [52]. We exclude the contributions of patch-based encoders from the evaluated metrics, as these models are substitutable and contribute equally across approaches. Additionally, we use SAM ViT-H for this analysis, as smaller SAM variants perform worse than the ViT-H-based approach [38]. Table 1 and Figure 3 present the results, which compare three key metrics:

**Parameter Count.** REN trained for DINO ViT-B/8 has $10.1\times$ fewer parameters than EfficientViT-SAM [51], the most parameter-efficient SAM-based model.

**Processing Time.** For a $32\times32$ grid of point prompts, REN generates region tokens $61.7\times$ faster than EfficientViT-SAM, the fastest SAM-based approach. Batched-SAM [24] was evaluated using a batch of 8 images—the largest it can handle without running out of memory on an A40. For fairness, REN was also benchmarked for a batch size of 8, though it can process a much larger batch of images.

**Peak GPU Memory.** While REN can process dense grids of point prompts simultaneously, the SAM-based methods split prompts into smaller batches and process them separately. On a single A40, SAM, EfficientViT-SAM, and SAM 2 cannot handle 4096 prompts at once, and Batched-SAM fails beyond this limit. In contrast, REN can process up to 16384 prompts ($64\times64$ grid) at once, while using less memory than what the most efficient SAM-based approach requires for just 64 prompts. For a $32\times32$ grid, REN uses $35.2\times$ less GPU memory than the most efficient SAM-based method.

### 4.2 Downstream Tasks

We compare REN's region representations to the original patch-based representations across several tasks. REN consistently outperforms both patch-based baselines and prior region-based methods, demonstrating its effectiveness and versatility.

**Visual Query Localization.** We evaluate REN on the Ego4D VQ2D benchmark, where the task is to localize the last occurrence of a query object in a long video. Our approach follows a stage-wise pipeline: (1) Use REN to extract region tokens from multi-resolution crops of video frames and the visual query; (2) use cosine similarity to select candidate regions in the video that match the query; (3) convert the point prompt of the selected candidates into bounding boxes using SAM 2 [35] and refine the selections by cropping around them; (4) use SAM 2 to track the last candidate as the initial

Table 2: **Visual query localization on the Ego4D VQ2D benchmark.** Our method substantially outperforms existing approaches, including those specifically developed for this task. Baseline results are sourced from the official leaderboard.

| Method | stAP | tAP | Succ. | Rec. |
|---|---|---|---|---|
| SiamRCNN [13] | 0.13 | 0.21 | 41.6 | 34.0 |
| CocoFormer [48] | 0.18 | 0.26 | 48.1 | 43.2 |
| VQLoC [18] | 0.24 | 0.32 | 55.9 | 45.1 |
| HERO-VQL | 0.28 | 0.37 | 60.7 | 45.3 |
| PRVQL [10] | 0.28 | 0.37 | 59.4 | 45.7 |
| RELOCATE [20] | 0.35 | 0.43 | 60.1 | **50.6** |
| REN$_{DINOv2}$ | **0.40** | **0.52** | **61.2** | 49.3 |

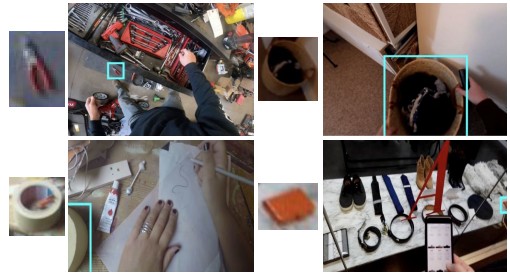

Figure 4: **Examples of query localization.** REN effectively localizes target objects in long videos despite challenges like clutter, occlusions, background blending, motion blur, viewpoint changes, and brief visibility.

Table 3: **Semantic segmentation using a linear classifier on frozen features.** For reference, the absolute state-of-the-art results from Chen et al. [6] and Wang et al. [44] are reported in the column heading. Results for methods using external segmenters are taken from Shlapentokh-Rothman et al. [38].

| Method | VOC2012 (89.0) | ADE20K (63.0) |
|---|---|---|
| *With External Segmenters* | | |
| SAM$_{DINOv2}$ | 83.6 | 50.2 |
| SAM+SLIC$_{DINOv2}$ | **86.9** | **52.9** |
| *Direct Encoders* | | |
| DINOv2 | 82.1 | 47.7 |
| REN$_{DINOv2}$ | **86.5** | **50.9** |
| DINO | 66.4 | 31.8 |
| REN$_{DINO}$ | **71.4** | **35.1** |
| OpenCLIP | 71.4 | 39.3 |
| REN$_{OpenCLIP}$ | **78.0** | **42.8** |

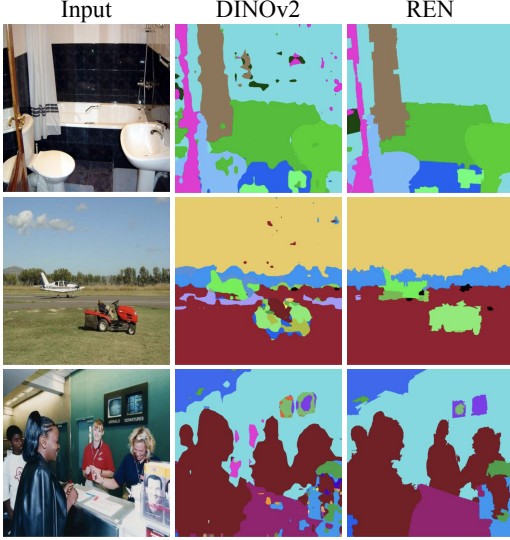

Input     DINOv2     REN

Figure 5: **Qualitative comparison of semantic segmentation on ADE20K.** Region tokens from REN yield cleaner, less noisy predictions compared to patch-based features from DINOv2.

track; and (5) use REN to generate additional visual queries from the initial track, and repeat the search, refinement, and tracking steps. This pipeline follows the approach proposed by Khosla et al. [20]. They provide a detailed explanation of each step. The main difference from [20]: they use SAM-based pooling to extract region tokens, while we use REN, and, given a $60\times$ speed-up realized by eliminating SAM, we can process multi-resolution crops of video frames while still being faster.

The results are presented in Table 2. REN significantly outperforms previous models, achieving a new state-of-the-art on this challenging task. Notably, the second-best method, RELOCATE, also uses a region-based approach that relies on SAM for region token generation. The strong performance of both RELOCATE and REN highlights the benefit of region-based representations for this task. Moreover, REN outperforms RELOCATE by leveraging a contrastive learning objective that enables it to capture fine-grained details of the query object. In contrast, RELOCATE generates region tokens by averaging features within object masks, which can sometimes overlook subtle distinguishing characteristics.

**Semantic Segmentation.** To evaluate representation quality, we follow the standard practice of training a linear classifier on frozen features to predict semantic class labels for objects in an image. The results are presented in Table 3. For each image encoder $E$, we compare two setups: (1) $E$: The encoder $E$ processes the input image to produce a low-resolution feature map. A linear classifier is applied to this feature map to predict class logits, which are upsampled to the original resolution for

Table 4: **Visual Haystacks' single-needle challenge.** Pooling SigLIP 2 features with a pretrained $REN_{DINO}$ leads to a substantial performance gain, outperforming proprietary LMMs, open-source LMMs, and RAG-based methods—especially at larger values of N. Results for all methods except REN and SigLIP 2 are from [46]. "E" indicates context overflow, execution failure, or API error.

| Method | N=1 | N=2 | N=3 | N=5 | N=10 | N=20 | N=50 | N=100 | N=500 | N=1K |
|---|---|---|---|---|---|---|---|---|---|---|
| Detector Oracle | 90.2 | 89.6 | 88.8 | 88.3 | 86.9 | 85.4 | 81.7 | 77.5 | 74.8 | 73.9 |
| Gemini 1.5 Pro | **88.4** | **82.0** | **78.3** | **76.0** | 71.9 | 68.6 | 62.8 | 57.4 | E | E |
| GPT-4o | 82.5 | 79.9 | 77.5 | 73.3 | 68.2 | 65.4 | 59.7 | 55.3 | E | E |
| LongVILA | 63.8 | 59.0 | 57.7 | 56.7 | 55.6 | 52.0 | 52.0 | 52.0 | E | E |
| Qwen2-VL | 80.9 | 76.6 | 73.6 | 67.9 | 62.6 | 59.1 | 52.6 | E | E | E |
| Phi-3 | 80.5 | 69.1 | 67.3 | 62.0 | 54.8 | 52.6 | 50.8 | E | E | E |
| InternVL2 | 88.1 | 80.5 | 72.3 | 63.9 | 58.8 | 55.2 | E | E | E | E |
| mPLUG-OWL3 | 84.4 | 66.0 | 62.1 | 57.0 | 53.2 | 51.5 | E | E | E | E |
| LLaVA-v1.5 | 85.8 | 77.1 | 75.8 | 68.6 | 63.6 | 60.4 | 55.3 | 57.5 | 55.4 | 52.9 |
| MIRAGE | 83.2 | 77.8 | 76.6 | 72.8 | 70.5 | 66.0 | 63.6 | 62.0 | 58.7 | 55.7 |
| SigLIP 2 | 72.0 | 69.2 | 68.1 | 65.3 | 64.1 | 60.3 | 58.7 | 58.3 | 56.6 | 54.9 |
| $REN_{DINO \cdot SigLIP2}$ | 81.2 | 78.6 | 77.4 | **76.0** | **74.0** | **72.1** | **68.3** | **65.5** | **62.3** | **59.2** |

Table 5: **One-shot image retrieval.** Baseline results are taken from Shlapentokh-Rothman et al. [38].

| Method | mAP | mRP@50 |
|---|---|---|
| DINOv2 | 0.13 | 0.33 |
| $SAM_{DINOv2}$ | 0.45 | 0.58 |
| $REN_{DINOv2}$ | **0.52** | **0.65** |

Query Image       Some Images Retrieved from the Database

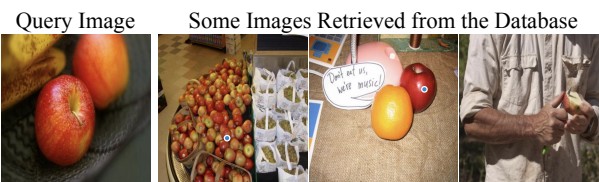

Figure 6: **Example of image retrieval on COCO.** Given an image of a query object (left), REN retrieves database images containing the same object.

per-pixel predictions; and (2) $REN_E$: REN generates region tokens for each point prompt on the input image, and a linear classifier predicts the class label for each token. Predictions are mapped back to the image by assigning the label to the patch/superpixel containing the corresponding prompt.

Table 3 also includes results from Shlapentokh-Rothman et al. [38], where external models are used to segment regions and a linear head predicts labels for each region. Using SAM for segmentation provides these methods with high boundary precision at substantial computational costs (see Section 4.1). Nevertheless, $REN_{DINOv2}$ outperforms $SAM_{DINOv2}$ due to better coverage of image regions.

**Finding Needle in a Haystack.** We evaluate our approach of extending pretrained REN to other image encoders (Section 3.3) on the single-needle challenge from the Visual Haystacks benchmark [46]. The task involves answering queries of the form: "*For the image with the [anchor object], is there a [target object]?*" We use $REN_{DINO}$ to segment each image into regions and employ SigLIP 2 [41] to generate text-aligned region tokens. Specifically, for each region identified by $REN_{DINO}$, we extract the corresponding patch features from the final hidden state of SigLIP 2's vision encoder and pool them using its pooling head to produce a region token. To answer a query, we first retrieve the image containing the anchor object by computing cosine similarity between region tokens of each image and the SigLIP 2 text embedding of "*This is a photo of [anchor object].*" The image with the most similar region token is selected. We then check whether this image contains the target object by computing cosine similarity between its region tokens and the text embedding for "*This is a photo of [target object].*" If the highest similarity exceeds 0.05, the answer is "Yes"; otherwise, "No."

Table 4 shows the results. While SigLIP 2 underperforms relative to the baselines, using $REN_{DINO}$ to pool its features into region tokens boosts the absolute performance by 8.7% on average. Furthermore, our approach remains robust as the number of input images (N) increases, and we outperform proprietary LMMs, open-source LMMs, and RAG-based methods reported in [46] by a substantial margin. This underscores the effectiveness of region-based representations in large-scale retrieval.

**Image Retrieval.** We evaluate REN on single-shot object-based image retrieval, where the task is to retrieve images from a database that contains a specified query object. Figure 6 illustrates an example.

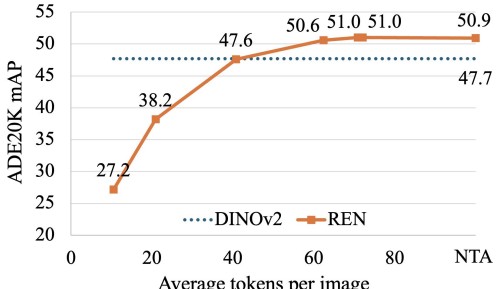

Figure 7: **Performance vs. token count.** REN matches the performance of the original image encoder, which uses 1369 tokens/image, with just 41 tokens/image, and surpasses it beyond that point. Performance stabilizes at 70 tokens/image. NTA denotes no token aggregation, i.e., 1369 tokens/image.

Table 6: **Effects of prompting technique and token aggregation.** SLIC-based prompting improves segmentation results on ADE20K, while image retrieval on COCO remains unaffected by the prompting strategy. Token aggregation reduces the average token count per image by $19.5\times$ without compromising performance. We use a similarity threshold of $\mu = 0.975$ for token aggregation, which yields an average of 72.1 tokens per image.

| Prompt | Token Aggregation | ADE20K (mAP) | COCO (mRP) |
|--------|-------------------|--------------|------------|
| Grid   |                   | 50.4         | 0.66       |
| Grid   | ✓                 | 50.4         | 0.65       |
| SLIC   |                   | 50.9         | 0.66       |
| SLIC   | ✓                 | 51.0         | 0.65       |

Table 7: **Training loss ablation.** A combination of contrastive loss and feature similarity loss yields the best performance. The VQ2D performance is evaluated on 500 videos from the validation set.

| Contrastive Loss | Feature Similarity | VOC2012 | VQ2D |
|------------------|--------------------|---------|------|
| ✓                |                    | 77.7    | 46.9 |
|                  | ✓                  | 86.0    | 49.7 |
| ✓                | ✓                  | **86.5**| **63.7** |

Following Shlapentokh-Rothman et al. [38], we use the COCO validation set as the image database, and 50 images with corresponding object masks serve as query instances for each object class. Query features are computed by averaging region tokens generated from 128 point prompts within each query mask. These are compared to region features of all database images using cosine similarity, and images are ranked by their highest similarity score. We report the mean Average Precision (mAP) and mean Retrieval Precision at 50 (mRP@50) in Table 5. Region-based methods outperform the patch-based baseline, and REN further surpasses the SAM-based baseline while offering faster and more efficient region token generation.

## 4.3 Ablations

We discuss the design decisions that lead to an effective region encoder network. All ablations are conducted using REN trained with DINOv2 as the image encoder.

**Effect of Token Aggregation.** As described in Section 3.2, the extent of token aggregation in REN can be controlled by the aggregation threshold $\mu$. Setting $\mu \geq 1$ disables aggregation, while lower values result in more aggressive merging, reducing the token count. Figure 7 shows semantic segmentation performance across different values of $\mu$ ranging from 0.875 to 0.975. The horizontal axis indicates the average number of tokens per image, and the vertical axis shows mAP. REN matches the performance of the patch-based encoder using just 41 tokens and surpasses it beyond that point. With $\mu = 0.975$, REN achieves a $19.5\times$ reduction in token count without compromising performance. Unlike patch-based encoders, which use a fixed token count (1369 tokens per image in this case), REN provides the flexibility to adjust the token count based on compute constraints.

**Effect of Prompting Strategies.** Table 6 compares grid-based and SLIC-based prompting strategies. The SLIC-based approach performs better for semantic segmentation because it results in more precise predictions at object boundaries, as illustrated in Figure 2. For image retrieval, both prompting strategies perform similarly. This is expected because noisy tokens at object boundaries in grid-based prompting are either attenuated during token aggregation or, if aggregation is not used, contribute minimally to the overall similarity scores between the query token and the rest of the image tokens.

**Training Loss Ablation.** Table 7 presents the results of a loss ablation study, showing that combining the contrastive loss and feature similarity loss yields the best performance.

# 5   Conclusion

In this work, we present REN, a model for efficiently generating region-based representations of an image. By removing the reliance on segmentation models, REN greatly improves the efficiency of region token extraction, making them more practical for a wide range of applications. Beyond efficiency, our learning-based approach produces region representations that not only surpass traditional patch-based features but also outperform prior region-based methods. Overall, our findings demonstrate that region tokens provide a compact, content-aware, and semantically rich alternative to patch-based representations, and REN offers a practical and efficient way of generating them.

**Limitations.** Using point prompts can introduce ambiguity—whether to represent a whole object or just a part. This is addressed by placing multiple prompts on the same object and adjusting the aggregation threshold $\mu$ based on the task: a high $\mu$ favors full objects, while a low $\mu$ favors parts. Additionally, region segmentations from REN are less precise than those from SAM, limiting its suitability for interactive segmentation, though they are sufficient for generating region tokens.

**Future Works.** Future work can explore using REN's region tokens as inputs to vision-language or multimodal models, potentially replacing patch-based tokens. Further directions include fine-tuning the image encoder with task objectives and explicitly modeling region relationships to better support complex scene understanding.

# Acknowledgements

This work is supported in part by ONR award N00014-23-1-2383, U.S. DARPA ECOLE Program No. #HR00112390060, National Science Foundation grants #2008387, #2045586, #2106825, and NIFA award 2020-67021-32799. We used GPUs at NCSA Delta through allocation CIS240541 from the Advanced Cyberinfrastructure Coordination Ecosystem: Services & Support (ACCESS) program, which is supported by U.S. National Science Foundation grants #2138259, #2138286, #2138307, #2137603, and #2138296. The views and conclusions contained herein are those of the authors and should not be interpreted as necessarily representing the official policies, either expressed or implied, of DARPA, ONR, or the U.S. Government.

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

## Supplementary Material

This supplementary material is organized as follows: Appendix A provides implementation details; Appendix B discusses alternative approaches explored during the development of REN; and Appendix C addresses broader societal impacts of our work.

## A    Implementation Details

We implement REN using PyTorch, with all training and evaluation performed on a single NVIDIA A40 GPU. For the vision backbone, we use frozen features from three pretrained encoders—DINO ViT-B/8, DINOv2 ViT-L/14, and OpenCLIP ViT-g/14—as detailed in Table 8. The region encoder comprises 4 cross-attention decoder layers with 8 attention heads.

**Training.** Training is performed on images sampled from the Ego4D dataset [13]. We use a batch size of 16 and randomly sample up to 256 point prompts per image to generate region tokens. Segmentation masks are precomputed for all training images using the SAM Automatic Mask Generator [22], with a $32\times32$ point grid and a stability threshold of 0.9. These masks are used to assign region IDs for the contrastive loss and to average-pool encoder features for the feature similarity loss. If multiple masks overlap a given point prompt, we select the mid-sized one.

To compute the contrastive loss, we create two augmented views of each image using a combination of horizontal flipping, color jittering, sharpness adjustment, affine transformations (rotation and shear), cropping, and resizing. Both the contrastive and feature similarity losses are weighted equally during training. The model is optimized using AdamW with a learning rate of 0.001, cosine decay schedule, 100 warmup steps, and gradient clipping with a maximum norm of 5.0.

**Inference.** For tasks where we do not extend a pretrained REN to a new image encoder, we use either grid-based or SLIC-based prompting. For SLIC-based prompting, we use Fast-SLIC [21] with a compactness value of 256. The number of point prompts is set to match the number of patches produced by the image encoder (e.g., 1369 for DINOv2). Token aggregation is then performed by constructing an adjacency graph using SciPy [19], where an edge is added between two region tokens if their cosine similarity exceeds a threshold $\mu$. Connected components are identified via breadth-first search (also using SciPy), and groups with fewer than three region tokens are discarded. Figure 8 reports the average number of region tokens per image for varying values of $\mu$; we find that $\mu = 0.975$ performs consistently well across tasks. Additional task-specific parameters are detailed in Section 4.

For tasks involving the extension of a pretrained REN to a new image encoder, we use SLIC-based prompting with 576 point prompts and a compactness of 256. Among the available models, the REN trained on DINO features performs best as a region segmenter, possibly due to its smaller patch size. For these transfer settings, we use a lower aggregation threshold of $\mu = 0.8$.

Table 9 summarizes the datasets used in this work.

## B    Explored Alternatives

To better understand and validate the design decisions in REN, we experimented with some alternative approaches. We discuss these explored alternatives below along with their empirical outcomes and limitations.

**Attention Supervision.**    We examine the effect of direct attention supervision during training, which would encourage each region token to aggregate information from its corresponding object.

Table 8: **Image encoders used for training REN.** Each encoder is listed with its architecture, input resolution, and output feature dimensionality used for region token extraction.

| Encoder | Architecture | Image Resolution | Feature Dimension |
|---|---|---|---|
| DINO [5] | ViT-B/8 | $384\times384$ | 768 |
| DINOv2 [32] | ViT-L/14 | $518\times518$ | 1024 |
| OpenCLIP [17] | ViT-g/14 | $224\times224$ | 1408 |

| Dataset | Task(s) | License |
|---------|---------|---------|
| Ego4D | Training, Visual Query Localization | MIT License |
| ADE20K | Semantic Segmentation | BSD-3-Clause License |
| VOC2012 | Semantic Segmentation | CC BY 2.5 |
| COCO | Visual Haystacks, Image Retrieval | CC BY 4.0 |

Table 9: **Summary of datasets used in this work.** All datasets used in this work are standard academic benchmarks and publicly available. Licensing information for each dataset is provided.

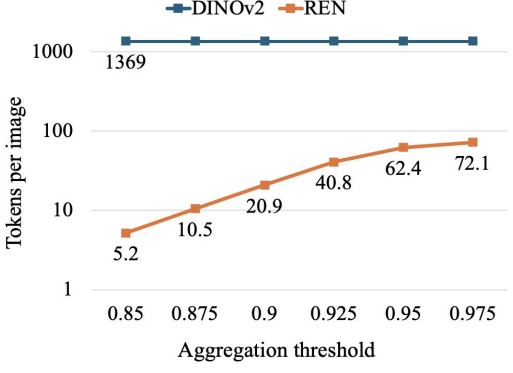

Figure 8: **Token count vs. aggregation threshold.** The average token count per image is computed using the ADE20K validation set.

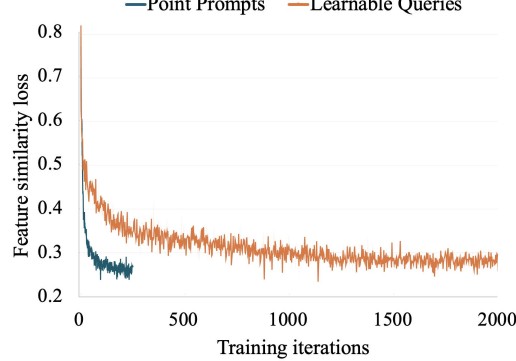

Figure 9: **Point prompts vs. learnable queries** Using point prompts leads to a faster convergence of feature similarity loss.

Specifically, for each point prompt, the model was trained to focus only on the relevant object by aligning its attention map with the object's SAM mask. Following Cheng et al. [7], we used a combination of binary cross-entropy and DICE loss, assigning equal weight to both components: $\mathcal{L}_{attn} = \mathcal{L}_{bce} + \mathcal{L}_{dice}$. In our experiments, applying attention supervision only to the attention maps of the final cross-attention block proved to be the most effective strategy for supervising attention maps. However, incorporating attention supervision ultimately degraded overall performance as shown in Table 10. We attribute this to the restrictive nature of direct attention supervision, which may hinder the model's ability to flexibly learn contextual representations and predict distinct representations for different point prompts within the same region.

**Point Prompts vs. Learnable Queries.** In the early stages of developing REN, we explored an alternative approach based on learnable queries, inspired by DETR [4] and MaskFormer [8]. In this setup, a fixed set of learnable queries was used to cross-attend patch features and produce region tokens. While this formulation yielded reasonable initial results, it introduced a complex optimization problem—primarily due to the use of Hungarian matching to align the predicted set of tokens with the target regions. In contrast, our current point-based formulation offers a significantly simpler and more efficient training pipeline. It eliminates the need for set matching and allows for faster convergence as shown in Figure 9. Moreover, point-based prompting offers greater flexibility and user control. For example, in tasks like Visual Query Localization and COCO image retrieval (Section 4), it is often desirable to generate region tokens for only a small subset of objects, such as a specific query object. The point-based approach makes this possible in a straightforward manner, which is difficult to achieve with fixed learnable queries.

## C  Broader Impact

REN's efficiency supports broader accessibility by enabling high-quality visual understanding on resource-constrained devices, such as mobile platforms or real-time systems in assistive technology and low-bandwidth settings. It also contributes to scientific progress by advancing foundational research in compact representation learning and enabling tasks like episodic memory retrieval. Furthermore, by eliminating the need to process hundreds of patch tokens or run large segmentation models, REN can help reduce the energy consumption and carbon footprint of large-scale video understanding systems.

Table 10: **Evaluating attention supervision as a training objective.** Incorporating attention supervision results in a performance drop for both semantic segmentation and visual query localization.

| Contrastive Loss | Feature Similarity | Attention Supervision | VOC2012 | VQ2D |
|:---:|:---:|:---:|:---:|:---:|
| ✓ | ✓ | | **86.5** | **63.7** |
| ✓ | ✓ | ✓ | 86.4 | 60.3 |

At the same time, the ability to efficiently localize and track objects in video streams could be misused for mass surveillance, particularly if deployed in real-time or without user consent. Additionally, incorrect predictions—such as false positives in object localization—could lead to harmful outcomes in safety-critical systems like autonomous vehicles or security monitoring. To mitigate these risks, we recommend incorporating safeguards such as transparency in deployment, human-in-the-loop oversight, and clear communication of failure modes. As foundational work, its societal impact will depend heavily on downstream use.

