# OpenReview forum: "REN: Fast and Efficient Region Encodings from Patch-Based Image Encoders"
_NeurIPS.cc/2025/Conference — NeurIPS 2025 poster_

### Official Review · Reviewer_PgVF · 2025-07-01

**Clarity:** 3
**Significance:** 3
**Originality:** 3
**Rating:** 4
**Confidence:** 3

**Summary:**

The paper introduces the Region Encoder Network (REN), a novel approach for generating efficient region-based image representations using point prompts. Traditional methods rely on class-agnostic segmenters (e.g., SAM) combined with patch-based encoders (e.g., DINO), which incur high computational costs due to segmentation. REN bypasses this bottleneck by directly generating region tokens through lightweight cross-attention modules, achieving ​​60× faster token generation​​ and ​​35× reduced memory usage​​ while improving token quality. Trained with DINO, DINOv2, and OpenCLIP encoders, REN demonstrates superior performance on semantic segmentation, retrieval, and challenging benchmarks like Ego4D VQ2D and Visual Haystacks.

**Questions:**

Please see the Weaknesses.

**Ethical Concerns:**

["NO or VERY MINOR ethics concerns only"]

**Final Justification:**

I think the authors have addressed my questions. I updated the rating to Borderline accept.

**Limitations:**

Yes

**Quality:**

3

**Strengths And Weaknesses:**

Strengths:

(1) Clear problem motivation: Highlights inefficiencies of patch-based methods and limitations of SAM-based region tokens.

(2) Well-structured methodology: Step-by-step explanation of REN’s architecture, training objectives (contrastive loss, feature similarity), and inference pipeline (prompting + aggregation).

Weaknesses:

(1) Architectural similarity to existing methods: The cross-attention module resembles DETR and MaskFormer, raising questions about novelty.

(2) Lacks formal proofs or guarantees for REN’s design choices (e.g., why cross-attention works for region tokenization).

(3) While REN reduces SAM’s computational cost, its performance still depends on SAM-generated masks for training and certain tasks (e.g., token aggregation thresholds).

(4) Limited exploration of alternatives: The ablation study (Section 4.3) focuses on prompting strategies but omits deeper comparisons with other attention variants (e.g., multi-head vs. single-head).

---

> ### Author Rebuttal · Authors · 2025-07-30
>
> Thank you for reviewing our work and providing constructive feedback. We are pleased to hear that you find our motivation clear and methodology well-structured. We have carefully addressed each of your comments and concerns below.
>
>
> > [Weakness 1] Architectural similarity to existing methods: The cross-attention module resembles DETR and MaskFormer, raising questions about novelty.
>
> REN differs fundamentally from DETR and MaskFormer in its goal, training paradigm, design, and usage. **DETR and MaskFormer are trained in a fully supervised manner for detection and segmentation**, respectively, whereas **REN is trained using self-supervised learning to produce efficient region-based image representations applicable across diverse tasks**, including semantic segmentation, instance-level localization, image retrieval, and text-to-image retrieval.
>
> Moreover, REN adopts a smaller and simpler cross-attention architecture, making it lighter and more efficient. Importantly, as discussed in Appendix B, **REN replaces fixed learnable queries with point prompts**, which has two key implications:
>
> (1) **Simplified training objective** – removing the need for Hungarian matching leads to significantly faster convergence (see Figure 9); and
>
> (2) **Greater flexibility and user control** – allowing region tokens to be generated for specific subsets of objects (e.g., a query object in Visual Query Localization or COCO retrieval), which is difficult to achieve with fixed learnable queries of DETR or Mask2Former.
>
> Thus, we think the fact that REN uses cross-attention does not diminish the methodological novelty of REN.
>
>
> > [Weakness 2] Lacks formal proofs or guarantees for REN’s design choices (e.g., why cross-attention works for region tokenization).
>
> While formal proofs are uncommon in deep learning, our design choices are validated through ablation studies and other experiments, which is an established and widely accepted practice.
>
>
> > [Weakness 3] While REN reduces SAM’s computational cost, its performance still depends on SAM-generated masks for training and certain tasks (e.g., token aggregation thresholds).
>
> REN’s performance and the quality of its region tokens depend primarily on the feature extractor and the attention weights learned during training, not directly on SAM. **SAM masks are used only to pool backbone features to generate target** region tokens during training. **REN does not rely on SAM masks during token aggregation or inference**.
>
> To further support that REN’s performance is independent of SAM, we highlight **two key findings** from our paper:
>
> (1)  Prior works (e.g., [1]) directly use SAM masks for generating region tokens, yet REN consistently outperforms them across a wide range of tasks (Section 4.2).
>
> (2)  In an experiment where SAM masks were used to directly supervise REN’s attention weights (see Appendix B), we observed reduced performance, indicating that REN benefits from learning its own attention rather than inheriting SAM’s segmentation.
>
> [1] Region-Based Representations Revisited. CVPR 2024.
>
>
> > [Weakness 4] Limited exploration of alternatives: The ablation study (Section 4.3) focuses on prompting strategies but omits deeper comparisons with other attention variants (e.g., multi-head vs. single-head).
>
> Beyond the ablation studies in Section 4.3, we explored the following alternative designs detailed in **Appendix B**:
>
> (1) **Supervising attention weights with SAM masks:** We tested whether directly supervising attention weights using SAM masks improves performance. The result was negative—this supervision slightly degraded performance.
>
> (2) **Adapting Mask2Former for region token generation:** We experimented with Mask2Former’s architecture and training objectives. While performance was comparable, this introduced unnecessary complexity, slowed training, and reduced user control.
>
> Other design choices, such as using multi-head self-attention, are well-established, have proven to be effective in prior works, and don’t really need further validation.

---

> > ### Author Response · Authors · 2025-08-05
> > **Rebuttal Follow-up**
> >
> > Dear Reviewer,
> >
> > We hope our rebuttal has addressed your comments and concerns. If so, we would greatly appreciate it if you could update your score accordingly. If there are any remaining issues, please let us know so we can address them promptly before the discussion period ends on August 8.
> >
> > Thank you for your time and consideration. We look forward to your response.
> >
> > Authors of REN

---

> > > ### Author Response · Authors · 2025-08-07
> > > **Rebuttal Follow-up**
> > >
> > > Dear Reviewer,
> > >
> > > We just wanted to kindly follow up on our earlier message. With the discussion period ending soon, we wanted to check if you had any further questions or concerns regarding our rebuttal. We’d be happy to clarify anything that remains unclear.
> > >
> > > Authors of REN

---

> > > > ### Author Response · Authors · 2025-08-08
> > > > **Rebuttal Follow-up**
> > > >
> > > > Dear Reviewer,
> > > >
> > > > As the discussion period closes in less than two days, we’d be very grateful if you could confirm whether our rebuttal addressed your concerns. Your response would help ensure a fair and informed decision. If any issues remain, we’d be happy to clarify them promptly.
> > > >
> > > > Authors of REN

---

### Official Review · Reviewer_Zqft · 2025-07-02

**Clarity:** 3
**Significance:** 2
**Originality:** 2
**Rating:** 4
**Confidence:** 4

**Summary:**

The paper introduces the Region Encoder Network (REN), a model that generates region-based image representations via point prompts. It bypasses expensive segmentation steps, using cross-attention to transform patch features into region tokens, achieving 60× faster token generation with 35× less memory. Experiments show REN outperforms patch-based encoders and SAM-based methods in efficiency and performance across tasks like semantic segmentation and retrieval.

**Questions:**

Please refer to the 'weakness' part.

**Ethical Concerns:**

["NO or VERY MINOR ethics concerns only"]

**Final Justification:**

Thanks for your responses, which have addressed most of my concerns. I will increase my score.

**Limitations:**

Please refer to the 'weakness' part.

**Paper Formatting Concerns:**

None.

**Quality:**

3

**Strengths And Weaknesses:**

Strengths:

1. The paper is well written and easy to follow.

2. The reduction in cost of the proposed method compared to SAM 2 is impressive.

3. The experimental results fully reflect the effectiveness of the proposed method.

Weaknesses:

1. The essence of REN's training is to establish the correlation between points, but the association between points and regions still seems to rely on other models for assistance, which restricts its application scope and makes it impossible to truly achieve real-time performance.

2. As described in lines 108-111 of Table 1, the RPN utilizes the prior information from SAM 2. I am curious whether this makes the training constitute an implicit distillation of SAM 2, which is why the RPN has performance advantages? Please replace the priors of other models or directly discuss this point.

3. Compared with DETR and Mask2Former, the RPN method is slightly less innovative, as it merely replaces learnable queries with point prompts. In addition, I am curious whether the region representations obtained by RPN are superior to those obtained by methods such as DETR and Mask2Former?

4. Can the proposed method be directly used for Fine-tuning like [1], and bring performance advantages to fine-grained tasks?


[1] UMG-CLIP: A Unified Multi-Granularity Vision Generalist for Open-World Understanding. ECCV 2024.

---

> ### Author Rebuttal · Authors · 2025-07-30
>
> Thank you for reviewing our work and providing constructive feedback. We appreciate your recognition of the paper’s clarity, the significant cost reduction, and the strong experimental results demonstrating the method’s effectiveness. We have carefully addressed each of your comments and concerns below.
>
>
> > [Weakness 1] The essence of REN's training is to establish the correlation between points, but the association between points and regions still seems to rely on other models for assistance, which restricts its application scope and makes it impossible to truly achieve real-time performance.
>
> As shown in Table 1, REN processes images at **more than 30 FPS**, demonstrating that it is **capable of real-time performance**. Importantly, **REN only uses external masks from SAM during training**. At inference, REN operates independently of any segmentation model—region tokens are generated directly from the backbone features and point prompts, making the method lightweight and efficient.
>
>
> > [Weakness 2] As described in lines 108-111 of Table 1, the RPN utilizes the prior information from SAM 2. I am curious whether this makes the training constitute an implicit distillation of SAM 2, which is why the RPN has performance advantages? Please replace the priors of other models or directly discuss this point.
>
> There is a typo in Section 3: we use SAM, not SAM 2, to generate region masks for training. This choice was made to ensure a fair comparison with baselines [1, 2], which also use SAM-generated masks. We appreciate you bringing this to our attention and we will correct it in the revision.
>
> [1]  Region-Based Representations Revisited. CVPR 2024.
>
> [2]  RELOCATE: A Simple Training-Free Baseline for Visual Query Localization Using Region-Based Representations. CVPR 2025.
>
>
> > [Weakness 3.1] Compared with DETR and Mask2Former, the RPN method is slightly less innovative, as it merely replaces learnable queries with point prompts.
>
> REN is fundamentally different from DETR and Mask2Former in its goal, training paradigm, design, and usage. **DETR and Mask2Former are trained in a fully supervised manner for detection and segmentation**, respectively. In contrast, **REN is trained using self-supervised learning to produce efficient region-based image representations applicable across diverse tasks**, including semantic segmentation, instance-level localization, image retrieval, and text-to-image retrieval. REN also employs a smaller and simpler cross-attention architecture, making it more lightweight and efficient.
>
> Furthermore, as discussed in Appendix B, using **point prompts instead of learnable queries has significant implications:**
>
> (1) **Simplified training objective** – removing the need for Hungarian matching leads to faster convergence (see Figure 9); and
>
> (2) **Greater flexibility and user control** – allowing region tokens to be generated for specific subsets of objects (e.g., a query object in Visual Query Localization or COCO retrieval), which is difficult to achieve with fixed learnable queries of DETR or Mask2Former.
>
>
> > [Weakness 3.2] In addition, I am curious whether the region representations obtained by RPN are superior to those obtained by methods such as DETR and Mask2Former?
>
> Unlike REN, **DETR and Mask2Former are not designed to produce general-purpose region-based representations**, but are specialized for detection and segmentation. As a result, REN’s region representations are more broadly applicable across diverse tasks (ref. Section 4.2).
> To further examine this, as discussed **in Appendix B, we adapted the Mask2Former architecture for region representation learning**. While it achieved performance comparable to REN, **it required significantly more complex optimization**, leading to slower training (see Figure 9). Moreover, its reliance on fixed learnable queries limits user control, making it **challenging to generate region representations for specific objects** within an image—a capability that REN’s point prompting naturally supports.
>
>
> > [Weakness 4] Can the proposed method be directly used for fine-tuning like [1], and bring performance advantages to fine-grained tasks?
> > [1] UMG-CLIP: A Unified Multi-Granularity Vision Generalist for Open-World Understanding. ECCV 2024.
>
> We think it’s possible; great suggestion for future work.

---

> > ### Author Response · Authors · 2025-08-05
> > **Rebuttal Follow-up**
> >
> > Dear Reviewer,
> >
> > We hope our rebuttal has addressed your comments and concerns. If so, we would greatly appreciate it if you could update your score accordingly. If there are any remaining issues, please let us know so we can address them promptly before the discussion period ends on August 8.
> >
> > Thank you for your time and consideration. We look forward to your response.
> >
> > Authors of REN

---

> ### Author Response · Authors · 2025-08-07
> **Rebuttal Follow-up**
>
> Dear Reviewer,
>
> We just wanted to kindly follow up on our earlier message. With the discussion period ending soon, we wanted to check if you had any further questions or concerns regarding our rebuttal. We’d be happy to clarify anything that remains unclear.
>
> Authors of REN

---

> > ### Author Response · Authors · 2025-08-08
> > **Rebuttal Follow-up**
> >
> > Dear Reviewer,
> >
> > As the discussion period closes in less than two days, we’d be very grateful if you could confirm whether our rebuttal addressed your concerns. Your response would help ensure a fair and informed decision. If any issues remain, we’d be happy to clarify them promptly.
> >
> > Authors of REN

---

### Official Review · Reviewer_Z7WG · 2025-07-02

**Clarity:** 3
**Significance:** 3
**Originality:** 3
**Rating:** 4
**Confidence:** 4

**Summary:**

To alleviate the computational burden of generating region-based representations by existing visual foundation models, this paper designs REN, a novel module that transforms patch-based image features into object-aligned region tokens using point prompts. Apart from the efficiency, REN also improves the quality of the generated region tokens under the supervision of mask generated by SAM2 while equipped with designed contrastive and feature similarity losses.

   Specifically, REN uses a lightweight stack of cross-attention blocks: point prompts attend to patch features from a frozen image encoder, producing tokens for every prompt. During training, REN is optimized with a contrastive loss (grouping tokens from the same object) and a feature-similarity loss (to align tokens with the encoder’s feature space), enabling it to “implicitly segment” objects through attention. At inference, REN can be prompted with a dense grid or superpixel-based points to cover the image.
By analyzing runtime and memory occupation, REN behaves more efficiently than other SAM-based methods. Experiments on several downstream tasks prove that REN is capable of extending to other encoders without dedicated training while achieving better performance.

**Questions:**

1. In Section Intro, “existing SAM-based methods for region token ...., which can remove fine-grained details and .... Current segmentation methods also .....”, how can you make sure that it’s the simple linear aggregation operation that causes the over-representation and no-representation phenomenon?
2. Fig. 1 can be redrawn to better epitomize the implementation details. The mask generated by SAM2 and the definition process of positive and negative samples should be added.
3. In Section 4.3, “all ablations are conducted using REN trained with DINOv2 as the image encoder.”, since DINOv2 with registers reports its segmentation improvement, have you ever tried to use DINOv2 with registers as your encoder?
4. Some descriptions in this paper are confusing. In Abs and Intro, this paper uses “region-based image representations”, “region representations”, “region tokens” and “region-based representations”. Are there any difference among these words? Or some of the words share the same meaning?

**Ethical Concerns:**

["NO or VERY MINOR ethics concerns only"]

**Final Justification:**

I think the authors have addressed my concerns. According to the contributions of the paper, I will keep my rating.

**Limitations:**

Yes

**Quality:**

3

**Strengths And Weaknesses:**

Strengths of the paper include comprehensive experiments that cover both Processing time comparisons and evaluation on promising downstream tasks, as well as the detailed analysis of strong performance of the proposed method. However, weaknesses are noted in the drawing of the method diagram. Additionally, the proposed method does not consider specific scenes, which means that what kind of scenes can make the best use of the proposed method.
Strength:
1. REN replaces an explicit segmentation step with cross-attention from point prompts, a creative alternative to methods that average-pool patch features within segmentation masks. The use of contrastive + feature-similarity losses to train these region tokens is a clever mechanism to ensure tokens remain semantically meaningful and aligned with the backbone features.
2. Region tokens generated through REN are very compact. The parameters and inference time, as well as the cuda memory occupation all get huge improvement according to the experiments. Besides, in the experiments on downstream tasks, REN consistently improves accuracy, showing its superiority.

Weakness:
1. Fig. 1 is the core diagram that presents the whole pipeline of the proposed method. However, this paper lacks some crucial components in Fig. 1, such as the mask generated by SAM2. Since the mask is the Supervision signal of the network, readers may feel confused because of its incomplete presentation.
2. This paper discusses images with few objects in Intro and argues it as a reason which causes the inefficient of current patch tokens generation. Although REN shows its efficiency in such images, it lacks of analysis about images with dense objects.

---

> ### Author Rebuttal · Authors · 2025-07-30
>
> Thank you for reviewing our work and providing constructive feedback. We appreciate your recognition of REN’s creative design, efficiency, compact region tokens, and strong performance validated by comprehensive experiments. We have carefully addressed each of your comments and concerns below.
>
> > [Weakness 1 & Question 2] Fig. 1 can be redrawn to better epitomize the implementation details. The mask generated by SAM2 and the definition process of positive and negative samples should be added.
>
> Thank you for the suggestion. We will update Figure 1 to include the missing training components—specifically, the SAM-generated masks, the target region tokens used for feature similarity loss, and the positive/negative samples used for contrastive learning.
>
>
> > [Weakness 2] This paper discusses images with few objects in Intro and argues it as a reason which causes the inefficient use of current patch tokens generation. Although REN shows its efficiency in such images, it lacks analysis about images with dense objects.
>
> Our argument in the intro is that the **number of tokens needed to represent an image should depend on its complexity**, which is not the case for current patch-based methods. For example, on Pascal VOC, we can represent images using fewer than 30 tokens on average while achieving performance comparable to a patch-based counterpart that uses 1,369 tokens per image. A similar analysis on ADE20K shows that we achieve comparable performance using 41 tokens per image instead of 1,369 (Figure 7). This demonstrates that **REN allocates more tokens as scene complexity increases** (30 → 41 on average), enabling efficient handling of both simple and dense scenes.
>
> Furthermore, REN effectively handles highly complex and dense images as well. This is evident from our results **on the challenging Ego4D VQ2D task**, which involves localizing small, overlapping, and occluded objects in cluttered scenes (see Figure 4). **REN significantly outperforms** all prior methods, including those specifically designed for this task.
>
>
> > [Question 1] In Section Intro, “existing SAM-based methods for region token ...., which can remove fine-grained details and .... Current segmentation methods also .....”, how can you make sure that it’s the simple linear aggregation operation that causes the over-representation and no-representation phenomenon?
>
> Simple linear aggregation of features treats all pixels equally, which can dilute important details—over-representing irrelevant regions and under-representing key discriminative areas. Additionally, restricting feature aggregation strictly to the object region can discard valuable contextual information. This effect is reflected in our results in Tables 7 and 10, summarized below:
>
> | Method                                   | Characteristic                        | VOC2012 | VQ2D  |
> |---------------------------------------|-------------------------------------|:-------:|:-----:|
> | Feature similarity only (Table 7)         | Learn to mimic linear aggregation of features            |  86.0   | 49.7  |
> | Attention supervision (Table 10)        | Restricts aggregation to object region with equal weighting |  86.4   | 60.3  |
> | Proposed (contrastive + feature similarity) | Learn discriminative feature aggregation | **86.5** | **63.7** |
>
> Furthermore, **REN consistently outperforms SAM-based methods—which rely on simple linear aggregation**—across a wide range of tasks (Section 4.2), reinforcing that learned discriminative aggregation leads to superior region representations.
>
>
> > [Question 3] In Section 4.3, “all ablations are conducted using REN trained with DINOv2 as the image encoder.”, since DINOv2 with registers reports its segmentation improvement, have you ever tried to use DINOv2 with registers as your encoder?
>
> We conduct additional ablations using DINOv2 with registers as the image encoder, focusing on semantic segmentation on the Pascal VOC validation set. The results are consistent with the ablations reported in the paper:
>
> **Training loss ablation:** A combination of contrastive and feature similarity loss performs the best.
>
> | Contrastive Loss | Feature Similarity | VOC2012 |
> |:----------------:|:-----------------:|:---------:|
> | ✓                |                     |  76.3     |
> |                    | ✓                 |  88.9     |
> | ✓                | ✓                 | **89.4** |
>
> **Effect of prompting strategy and token aggregation:** SLIC-based prompting provides slightly better performance, and token aggregation maintains accuracy while reducing the number of tokens required to represent each image.
>
> | Prompt | Token Aggregation | VOC2012 |
> |:----------------:|:-----------------:|:-------:|
> | Grid                |                      |  89.1 |
> | Grid                | ✓                  |  89.1 |
> | SLIC               |                      |  89.3 |
> | SLIC                | ✓                 |  89.4 |
>
> As expected, using DINOv2 with registers provides a marginal performance improvement:
>
> | Image Encoder  | mIoU |
> |----------------|:-----------------:|
> | DINOv2                              | 88.9     |
> | DINOv2 with registers        | 89.4     |
>
> These results show that **REN’s performance scales with backbone improvements for the given task while maintaining its core performance trends and ablation conclusions**.
>
>
> > [Question 4] Some descriptions in this paper are confusing. In Abs and Intro, this paper uses “region-based image representations”, “region representations”, “region tokens” and “region-based representations”. Are there any differences among these words? Or some of the words share the same meaning?
>
> All these phrases mean the same thing. We will clarify this better in the revision.

---

> > ### Author Response · Authors · 2025-08-05
> > **Rebuttal Follow-up**
> >
> > Dear Reviewer,
> >
> > We hope our rebuttal has addressed your comments and concerns. If so, we would greatly appreciate it if you could update your score accordingly. If there are any remaining issues, please let us know so we can address them promptly before the discussion period ends on August 8.
> >
> > Thank you for your time and consideration. We look forward to your response.
> >
> > Authors of REN

---

> > > ### Author Response · Authors · 2025-08-07
> > > **Rebuttal Follow-up**
> > >
> > > Dear Reviewer,
> > >
> > > We just wanted to kindly follow up on our earlier message. With the discussion period ending soon, we wanted to check if you had any further questions or concerns regarding our rebuttal. We’d be happy to clarify anything that remains unclear.
> > >
> > > Authors of REN

---

> > > > ### Author Response · Authors · 2025-08-08
> > > > **Rebuttal Follow-Up**
> > > >
> > > > Dear Reviewer,
> > > >
> > > > As the discussion period closes in less than two days, we’d be very grateful if you could confirm whether our rebuttal addressed your concerns. Your response would help ensure a fair and informed decision. If any issues remain, we’d be happy to clarify them promptly.
> > > >
> > > > Authors of REN

---

> > ### Comment · Reviewer_Z7WG · 2025-08-08
> >
> > Thanks for the response to my concerns.  I think the authors have addressed my concerns. According to the contributions of the paper, I will keep my rating.

---

### Official Review · Reviewer_TN4E · 2025-07-02

**Clarity:** 3
**Significance:** 2
**Originality:** 2
**Rating:** 4
**Confidence:** 3

**Summary:**

* This paper introduces the Region Encoder Network (REN), a model that efficiently generates region-level representations from patch-based image encoders such as DINO, DINOv2, and OpenCLIP.
* REN directly produces region tokens by leveraging point prompts as queries and applying cross-attention over patch features, achieving a 60× speedup and 35× lower memory usage compared to class-agnostic segmenter (e.g., SAM) based approaches.
* REN is trained with a combination of contrastive loss and feature similarity loss, which enhances object-level discriminative power while preserving the generalization capability of the backbone.
* Across various benchmarks (semantic segmentation, visual query localization, and image retrieval), REN consistently outperforms or matches patch-based and SAM-based region representations with higher efficiency.

**Questions:**

Please refer to the weaknesses.

**Ethical Concerns:**

["NO or VERY MINOR ethics concerns only"]

**Final Justification:**

The authors have addressed several concerns, including clarifying the “segmentation-free” definition, showing that SAM dependency is limited to training, adding quantitative evidence for feature similarity loss, and demonstrating robustness in dense and complex scenes.   They also provided results on stronger backbones and clarified differences from DETR and Mask2Former.
These efforts are appreciated, and for these reasons, I will maintain my original borderline accept rating.

**Limitations:**

yes

**Quality:**

2

**Strengths And Weaknesses:**

## **Strengths**

* REN eliminates the primary bottleneck of region token generation by removing the segmentation step and employs a lightweight cross-attention structure, resulting in substantial improvements in speed (up to 60× faster) and memory efficiency (up to 35× lower usage) compared to SAM-based methods.


* The method is experimentally validated to work with various patch-based image encoders, including DINO, DINOv2, and OpenCLIP, and can be extended to new encoders without requiring additional training.


* REN demonstrates superior performance over patch-based and SAM-based region representation methods on multiple benchmarks, including semantic segmentation, visual query localization (Ego4D VQ2D), and Visual Haystacks.


## **Weaknesses**

* While REN is described as "segmentation-free," the training process relies on pseudo-labels generated by SAM, implying an indirect dependency on segmentation and possible interpretation as knowledge distillation from SAM.
* Although the authors emphasize the importance of preserving alignment with the original encoder via feature similarity loss, the ablation study is limited and lacks quantitative analysis on its impact on downstream tasks.
* The self-supervised region token learning in REN, based on contrastive learning for object-level consistency, can be viewed as an incremental extension of prior works such as TokenCut and LOST, rather than a fundamentally novel paradigm.
* As acknowledged by the authors, there exists inherent ambiguity regarding whether a single point prompt represents the entire object or only a part, requiring heuristic adjustment of the aggregation threshold (µ) based on the task.
  * The point prompting strategy may lead to reduced accuracy in complex scenes containing overlapping objects, where precise region separation is more challenging.
* As acknowledged by the authors, REN's region segmentation provides lower boundary precision compared to SAM, making it less suitable for applications requiring high-precision interactive segmentation.
* Minor Typo: Line 303, "limiting it’s" → "limiting its".

---

> ### Author Rebuttal · Authors · 2025-07-30
>
> Thank you for reviewing our work and providing constructive feedback. We appreciate your recognition of REN’s efficiency, broad encoder compatibility, and strong performance across multiple benchmarks. We have carefully addressed each of your comments and concerns below.
>
>
> > [Weakness 1] While REN is described as "segmentation-free," the training process relies on pseudo-labels generated by SAM, implying an indirect dependency on segmentation and possible interpretation as knowledge distillation from SAM.
>
> When we describe REN as “segmentation-free,” we refer to its inference phase: **region tokens are generated directly from the trained REN model without performing any explicit segmentation** step, unlike prior works. While pseudo-labels from SAM are used only during training to guide learning, this does not impose a segmentation requirement at inference. For instance, all downstream tasks in Section 4.2 operate without any segmentation step, resulting in significantly lower computational time and memory usage.
>
>
> > [Weakness 2] Although the authors emphasize the importance of preserving alignment with the original encoder via feature similarity loss, the ablation study is limited and lacks quantitative analysis of its impact on downstream tasks.
>
> The ablation study presented in **Table 7** (copied below for quick reference) **already demonstrates that** using both feature similarity and contrastive loss yields the best overall performance. To further quantify the benefit of feature alignment, we additionally report results on the **COCO retrieval task: the mRP is 0.56 when only contrastive loss is used, and it is 0.65 when feature similarity loss is also used** (higher is better). This now constitutes an ablation across all downstream tasks discussed in the paper, except for Visual Haystacks (because there REN is used purely as a segmentation model to guide feature pooling from SigLIP 2 features).
>
> | Contrastive Loss | Feature Similarity | VOC2012 | VQ2D  |
> |:----------------:|:-----------------:|:-------:|:-----:|
> | ✓                |                   |  77.7   | 46.9  |
> |                  | ✓                 |  86.0   | 49.7  |
> | ✓                | ✓                 | **86.5**| **63.7** |
>
>
> > [Weakness 3] The self-supervised region token learning in REN, based on contrastive learning for object-level consistency, can be viewed as an incremental extension of prior works such as TokenCut and LOST, rather than a fundamentally novel paradigm.
>
> We respectfully disagree. **TokenCut and LOST are training-free methods designed specifically for object localization** in images. In contrast, **REN is a trained model that learns generalizable image representations applicable to a wide range of tasks**, including semantic segmentation, instance-level localization, image-based retrieval, and text-based retrieval. The only commonality is that both approaches leverage object-level cues present in patch features from self-supervised Transformer models (as noted in lines 81–83), but **REN’s goal, training paradigm, and capabilities differ fundamentally from TokenCut and LOST**.
>
>
> > [Weakness 4.1] As acknowledged by the authors, there exists inherent ambiguity regarding whether a single point prompt represents the entire object or only a part, requiring heuristic adjustment of the aggregation threshold (µ) based on the task.
>
> We have acknowledged this as a limitation. However, as shown in Figure 7, REN’s performance is robust across a wide range of aggregation thresholds (µ), indicating that the model is not highly sensitive to this parameter in practice.
>
>
> > [Weakness 4.2] The point prompting strategy may lead to reduced accuracy in complex scenes containing overlapping objects, where precise region separation is more challenging.
>
> We think the point prompting strategy is more adaptive and better suited for complex scenes, as the **number of point prompts can be increased to capture finer object details** in dense regions. This advantage is evident in our results on the challenging Ego4D VQ2D task, which involves localizing small, overlapping, and occluded objects in highly cluttered scenes (see Figure 4). **REN significantly outperforms prior methods**, including those specifically designed for this task, highlighting its effectiveness in handling overlapping objects and complex layouts. Moreover, our segmentation results on ADE20K, where REN surpasses patch-based baselines, further support this conclusion.
>
>
> > [Weakness 6] Minor Typo: Line 303, "limiting it’s" → "limiting its".
>
> Thanks for pointing it out; we will fix it in the revision.

---

> > ### Author Response · Authors · 2025-08-05
> > **Rebuttal Follow-up**
> >
> > Dear Reviewer,
> >
> > We hope our rebuttal has addressed your comments and concerns. If so, we would greatly appreciate it if you could update your score accordingly. If there are any remaining issues, please let us know so we can address them promptly before the discussion period ends on August 8.
> >
> > Thank you for your time and consideration. We look forward to your response.
> >
> > Authors of REN

---

> > > ### Author Response · Authors · 2025-08-07
> > > **Rebuttal Follow-up**
> > >
> > > Dear Reviewer,
> > >
> > > We just wanted to kindly follow up on our earlier message. With the discussion period ending soon, we wanted to check if you had any further questions or concerns regarding our rebuttal. We’d be happy to clarify anything that remains unclear.
> > >
> > > Authors of REN

---

> > ### Comment · Reviewer_TN4E · 2025-08-08
> >
> > Thank you for your detailed and thoughtful responses.
> > It appears that many of my earlier concerns have been addressed.
> > I will take your responses into account, together with the weaknesses raised by other reviewers, to make my final decision.

---

> ### Author Response · Authors · 2025-08-08
> **Rebuttal Follow-up**
>
> Dear Reviewer,
>
> As the discussion period closes in less than two days, we’d be very grateful if you could confirm whether our rebuttal addressed your concerns. Your response would help ensure a fair and informed decision. If any issues remain, we’d be happy to clarify them promptly.
>
> Authors of REN

---

### Author Response · Authors · 2025-08-08
**Rebuttal Summary and Request for Further Feedback**

Dear Reviewers and Area Chair,

We understand the time constraints and busy schedules during the review process, so to make it easier to follow our responses, we’ve included a concise, point-by-point summary of the main clarifications in our rebuttal. Each point is annotated with the corresponding reviewer ID for reference. For detailed justifications and experimental results, please refer to the full response under the respective comment. We hope this summary provides a clear high-level overview of our rebuttal at a quick glance.

**Segmentation-Free Claim (Reviewer TN4E, Zqft)**
- REN is **segmentation-free at inference**, which enables a 60x faster token generation and 35x less memory requirement; pseudo-labels from SAM are used only during training.
- No segmentation step is involved in any downstream task presented in Section 4.

**Architectural Novelty vs. DETR/Mask2Former (Reviewers Zqft, PgVF)**
- **Objective/Design/Use:** REN is developed for general purpose representation learning; DETR/Mask2Former are designed for detection/segmentation.
- **Supervision:** REN is self-supervised and general-purpose; DETR/Mask2Former are supervised.
- **Architecture:** REN uses point prompts for cross-attention with image features; DETR/Mask2Former use learnable queries. This change leads to significantly faster training (Figure 9) and an option to generate representations for only specific objects in an image (see Appendix B).
- **Training:** REN *does not* use Hungarian matching, which significantly simplifies the training objective (see Figure 9).
- Only similarity is that they all use cross-attention—a common mechanism used in many other works.

**Comparison with TokenCut/LOST (Reviewer TN4E)**
- REN is a **trained representation learner applicable across diverse tasks** (e.g., retrieval,  segmentation); TokenCut and LOST are **training-free methods designed only for object localization**.
- REN’s goal, capabilities, and design are fundamentally different.

**Efficiency and Real-Time Performance (Reviewer Zqft)**
- REN runs at **30+ FPS**, confirming real-time capability.
- SAM is used only during training; inference is segmentation-free and fast.

**Handling Complex and Dense Scenes (Reviewer Z7WG, TN4E)**
- REN **dynamically adjusts token count based on image complexity** (e.g., on average 30 tokens/image on VOC vs. 41/image on ADE20K).
- **State-of-the-art results on Ego4D VQ2D**, involving overlapping and occluded objects, demonstrate robustness in dense scenes.

**Feature Similarity Loss Justification (Reviewer TN4E)**
- Additional results (e.g., COCO retrieval mRP improves from 0.56 to 0.65) confirm the importance of feature similarity loss.
- This complements the existing ablation **already presented in Table 7 of the paper**.

**Comparing REN's Feature Aggregation to Linear Aggregation (Reviewer Z7WG)**
- Linear aggregation (used by SAM-based methods) treats all pixels equally and loses important details.
- Learned **attention-based aggregation** in REN achieves **superior performance (see ablations in Tables 7 & 10)**.
- REN consistently **outperforms approaches that use linear aggregation** (see comparison with SAM_X in Section 4.2).

**Ablation Scope and Alternatives (Reviewer PgVF)**
- **Appendix B** covers additional design variants: (1) Direct supervision of attention weights with SAM degrades performance. (2) Mask2Former adaptation is complex and less flexible.
- Other architectural choices (e.g., multi-head attention) follow standard practice.

---

### Decision · Program_Chairs · 2025-09-17

**Decision:**

Accept (poster)

**Comment:**

(a) This paper introduces the Region Encoder Network (REN), a novel module designed to efficiently generate high-quality, region-level image representations from patch-based encoders like DINO and OpenCLIP. The central scientific claim is that it is possible to bypass the computationally expensive explicit segmentation step (e.g., using SAM) common in prior work. REN achieves this by using point prompts as queries in a lightweight cross-attention mechanism over patch features, directly producing region tokens. The model is trained with a combination of a contrastive loss to ensure object-level consistency and a feature similarity loss to maintain alignment with the original encoder's feature space. The key findings are that REN is massively more efficient—achieving up to a 60x speedup and 35x memory reduction—while consistently outperforming or matching both patch-based and SAM-based region representation methods across a variety of downstream tasks like semantic segmentation, visual query localization, and image retrieval.

(b) The paper's most significant strength is the massive improvement in efficiency it offers for generating region tokens, directly addressing a major bottleneck in a practical and important area of computer vision. The 60x speedup and 35x memory reduction are impressive and highly impactful. The proposed REN method is creative, replacing a heavy segmentation step with a lightweight and clever cross-attention mechanism. Reviewers also praised the comprehensive experimental validation, which demonstrates REN's superiority and versatility across multiple encoders (DINO, DINOv2, OpenCLIP) and a wide range of diverse downstream tasks. The method is well-structured, clearly explained, and the results support the claims of the paper.

(c) Initial weaknesses centered on the method's novelty and its relationship to prior work. Some reviewers noted architectural similarities to DETR and Mask2Former, questioning the originality of the cross-attention module. There was also confusion about the "segmentation-free" claim, as the model relies on SAM-generated masks for its training supervision, implying an indirect dependency. Other concerns included a lack of certain ablations, questions about performance in dense scenes with overlapping objects, and the ambiguity of using single point prompts to represent entire objects which the authors have rebutted applies only to the inference stage.

(d) The recommendation is to Accept this paper. The primary reason is its clear, substantial, and highly practical contribution: a method that drastically reduces the computational cost of generating region-based representations while simultaneously improving their quality. The 60x speedup is a significant improvement enabling efficient runtime inference. The authors' rebuttal was also effective, successfully clarifying the method's novelty by distinguishing its self-supervised, representation-focused goal from the fully-supervised detection/segmentation goals of DETR and Mask2Former. They also convincingly addressed concerns about the dependency on SAM (clarifying it's for training only) and demonstrated the method's effectiveness in complex scenes. Unfortunately, two reviewers (PgVF
And Zqft) havent been involved in the discussion phase despite repeated requests and I am basing this on the author;s rebuttal and the key contribution of the paper.

(e) The rebuttal period was elaborate from the authors, with them given detailed responses addressing the main concerns of the reviewers A key point of discussion was the method's novelty relative to architectures like DETR and Mask2Former, raised by Reviewers Zqft and PgVF. The authors clarified this by explaining that REN's goal is fundamentally different (learning general-purpose representations via self-supervision vs. end-to-end supervised detection), and that their use of point prompts instead of learnable queries simplifies training and adds flexibility. This clarification was accepted by both the reviewers. The authors also addressed the confusion around the "segmentation-free" claim from Reviewer TN4E by explaining it applies to the inference stage, which is the source of the efficiency gains. Furthermore, they provided additional quantitative evidence and new experiments on stronger backbones to address other concerns, such as the impact of their feature similarity loss and performance in dense scenes, solidifying the reviewers' confidence in the work's contributions.